# Development of a revised Jalowiec Coping Scale for use by emergency clinicians: a cross-sectional scale development study

Jaimi H Greenslade [1,2] Marianne C Wallis,[3] Amy Johnston,[4,5] Eric Carlström,[6] Daniel Wilhelms,[7,8] Ogilvie Thom,[9] Louisa Abraham,[2] Julia Crilly,[10,11] The WES investigators

For numbered affiliations see end of article.

**Correspondence to**
A/Prof Jaimi H Greenslade; j.greenslade@qut.edu.au

## ABSTRACT

**Objectives** The aim of this study was to develop and validate a scale to measure the coping strategies used by emergency staff in response to workplace stress. To achieve this aim, we developed a refined Jalowiec Coping Scale (JCS), termed the Jalowiec Coping Scale-Emergency Department (JCS-ED) and validated this scale on a sample of emergency clinicians.

**Design** A cross-sectional survey incorporating the JCS, the working environment scale-10 and a measure of workplace stressors was administered between July 2016 and June 2017. The JCS-ED was developed in three stages: 1) item reduction through content matter experts, 2) exploratory factor analysis for further item reduction and to identify the factor structure of the revised scale and 3) confirmatory factor analyses to confirm the factors identified within the exploratory factor analysis.

**Setting** Six Emergency Departments (EDs) in Australia and four in Sweden. There were three tertiary hospitals, five large urban hospitals and two small urban hospitals.

**Participants** Participants were eligible for inclusion if they worked full-time or part-time as medical or nursing staff in the study EDs. The median age of participants was 35 years (IQR: 28–45 years) and they had been working in the ED for a median of 5 years (IQR: 2–10 years). 79% were females and 76% were nurses.

**Results** A total of 875 ED staff completed the survey (response rate 51%). The content matter experts reduced the 60-item scale to 32 items. Exploratory factor analyses then further reduced the scale to 18 items assessing three categories of coping: problem-focussed coping, positive emotion-focussed coping and negative emotion-focussed coping. Confirmatory factor analysis supported this three-factor structure. Negative coping strategies were associated with poor perceptions of the work environment and higher ratings of stress.

**Conclusions** The JCS-ED assesses maladaptive coping strategies along with problem-focussed and emotion-focussed coping styles. It is a short instrument that is likely to be useful in measuring the types of coping strategies employed by staff.

## Strengths and limitations of this study

► This study uses a large cohort of nurses and physicians from 10 emergency departments (EDs) in two countries.
► This coping scale has been validated for use in the ED and may also be relevant to the broader range of staff employed in an acute care setting.
► The Jalowiec Coping Scale-ED is practical as it is short enough to encourage inclusion in future surveys exploring ED staff perceptions.
► No longitudinal data were available, meaning that test–retest reliability was unable to be determined.
► The response rate was reasonable for a survey design; however, it is unknown whether there was non-response bias.

## INTRODUCTION

The emergency department (ED) is a dynamic and demanding environment. Positive factors such as a supportive team environment, the development of high-level clinical skills and the challenges around dealing with varied patient groups attract clinical staff to this setting.[1] However, high workload, inadequate staffing, workplace violence and dealing with critically ill patients can place strain on employees.[2] Indeed, high levels of psychological distress[3] and staff turnover[4] are evident among ED staff. The coping strategies employed by clinicians may be one critical component in distinguishing these contrasting views of the ED as a stressful vs positive environment.

Coping is defined as the thoughts and behaviours used to manage the demands of situations that are appraised as stressful.[5] The transactional model of stress and coping outlines the processes by which stressful situations give rise to coping behaviours, and ultimately to workplace well-being outcomes.[5]

Specifically, this model posits that individuals appraise or evaluate their situation. If a situation is deemed taxing or overwhelming, they will engage in thoughts or actions (coping strategies) to manage that situation.[6] [7] Many different coping strategies can be employed, and these depend on both the environment and on personal disposition.[8] Such coping strategies serve to alter stress in various ways. First, they may address the problem causing distress (problem-focussed coping).[8] Problem-focussed coping attempts to remove the source of distress, thereby removing the stressor.[6] As such, individuals who successfully use problem-focussed coping experience lower distress and, thus, experience fewer negative outcomes from stress.[6] Second, coping strategies may attempt to ameliorate the negative emotions associated with the stressor (emotion-focussed coping).[8] Emotion-focussed coping can change the way we think about or interpret what is happening.[6] Some emotion focussed coping strategies may be positive and functional, while others can have negative consequences.[8] For example, drinking to cope may provide short term relief from stress, but does not actually reduce the problem in the longer term. Such maladaptive coping does not reduce the impact of the stressful situation and, thus, is linked to poorer outcomes.[8] In line with this theory, previous studies have suggested that ED staff using problem-focussed coping have lower levels of burnout[9] and better psychological health[3] than those using emotion-oriented coping or maladaptive coping strategies. Identifying the functional and maladaptive coping strategies that are used within the ED is important as these are amenable to intervention, with therapies such as cognitive behaviour therapy being shown to successfully increase the use of beneficial strategies.[8]

One major challenge in identifying and modifying coping strategies within the ED is that the measurement of coping has been difficult. There are no validated scales of which we are aware that were designed to measure coping within the ED. As such, various generic coping scales have been applied to this setting.[9–12] Existing scales are lengthy,[8] or have displayed poor psychometric properties.[13] For example, the Coping Orientation to Problems Experienced (COPE) questionnaire is the most commonly used instrument in adult samples.[14] This questionnaire is a 53-item index, but has been found to have an equivocal factor structure with moderate reliability.[14] [15] The Jalowiec Coping Scale (JCS) is another coping scale, with the advantage that it was specifically developed to measure the process of coping in a healthcare setting. This scale incorporates 60 items distributed across eight dimensions and has been translated into >20 languages.[16] The JCS was originally developed to measure problem-focussed and emotion-focussed coping styles, but the final version identifies eight different dimensions of coping. The JCS has been used nationally and internationally with both patients and employees in the healthcare setting.[17] While this scale has been extensively used, studies assessing its psychometric properties have not

supported the 8-item structure.[18] Furthermore, a 60-item scale is arguably too long for busy ED staff.

The overarching goal of this study was to produce a coping scale that would be useful for assessing coping in ED staff. This instrument should incorporate items to identify beneficial and maladaptive problem-focussed and emotion-focussed coping strategies while also being short enough for easy completion in a busy work environment. To achieve this goal, we sought to develop and validate a modified JCS coping scale for the ED setting.

## METHODS
### Participants and design
All full-time and part-time medical and nursing staff employed in the study EDs were eligible for inclusion in this study. There were no exclusion criteria. A cross-sectional paper survey was administered between July 2016 and June 2017 depending on logistics at each site. Surveys were hand distributed to 1709 staff by a local investigator at each site, with 876 returned. One survey was excluded due to extensive missing data, making the final sample size 875 (51% response rate). Staff were also provided with information and invited to participate via email and in ward-based information sessions. No compensation was provided to staff for completing the survey. Surveys were returned to locked boxes within each hospital ED, or via stamped self-addressed envelopes. A reminder email was sent out 2 weeks after survey distribution.

### Instrument
The JCS was used to assess the coping strategies employed by ED clinicians. The JCS incorporates 60 items that were developed to measure eight different coping dimensions. These include facing up to the problem (confrontive coping), avoiding the problem (evasive coping), positive thinking (optimistic coping), pessimistic thinking (fatalistic coping), releasing emotions (emotive coping), making yourself feel better (palliative coping), using support systems (supportive coping) and depending on yourself (self-reliant coping). Respondents report how often they have used each coping method, with scores for each item ranging from 0 (*never used*) to 3 (*often used*).

The questionnaire also included the working environment scale-10 (WES-10) scale[19] and a measure of workplace stressors.[20] The WES-10 is a 10-item scale that describes four aspects of the working environment; opportunity for personal and professional growth (self-realisation, 4 items); workload (2 items); interpersonal conflict (2 items) and nervousness (2 items).[19] Respondents are asked to answer how they feel about each item on 1–5 scale, with Likert-scale labels differing according to the question asked. This scale has been used within an ED,[21] but was not specifically developed or validated in this setting. Cronbach's alpha for self-realisation was 0.72, while Spearman-Brown was 0.64 for workload, 0.57 for conflict and 0.70 for nervousness. These show moderate internal consistency and are in line with reliability

coefficients reported in previous studies.[19 21] Job stressors were assessed using a 15-item ED stressor scale (EDSS).[20] This scale was designed to assesses stressors reported by nurses within an Australian ED.[20] Respondents were asked to rate on a scale of 1–15, how stressful they would find each of 15 stress-provoking events. They also were asked how often they experienced each event, ranging from 0 (*never*) to 3 (*daily*). Items on this scale are not combined to form sub-scales, they each assess a different stressor within the ED.

### Patient and public involvement

This study did not include patients. As such, patients were not involved in the design or conduct of the study. Results will not be disseminated to patients directly.

### Data analysis

Statistical analyses were performed in R V.3.5.1.[22] Characteristics of responders were reported by site. A shortened JCS was then developed in three steps. First, item reduction was conducted through review and consensus by content matter experts. Second, exploratory factor analysis (EFA) was conducted to reduce the number of items and to identify the factor structure of the revised scales. For this step, principal axis factoring with oblimin rotation was estimated using the R psych package.[23] As the data were ordinal, factor analysis was based on the polychoric correlation matrix.[24] Third, once the final items and factor structure were identified through EFA, confirmatory factor analysis (CFA) was conducted. CFA is a method of construct validation that is used when there is theory or research that posits an underlying latent variable structure.[25] All items comprising a particular subscale are loaded onto their related factor, and the acceptability of this model is evaluated by goodness-of-fit measures and the strength of resulting parameter estimates.[25] CFA was conducted in the lavaan R package.[26]

After the modified scale was developed, descriptive statistics for each of the scales were reported by country. Moreover, the criterion validity of the scale was established. Criterion validity describes how well scores on one measure predict scores on another related measure.[27] Based on previous research, it was anticipated that maladaptive coping strategies would be associated with poorer perceptions of the work environment and high stress, while problem-focussed strategies would be associated with higher perceptions of the work environment and low stress. As such, the relationship between JCS subscales, the WES-10 subscales and the EDSS were examined. For the EDSS, we focussed on the five stressors that are most commonly reported in the ED: heavy workload, environmental concerns (eg, overcrowding), inability to provide optimal care, high acuity patients and workplace violence.[28 29] Scatter plots showed that all associations were linear with no influential outliers. However, intraclass correlations (ICCs) for several of the items were >5%, indicating site-level variance in these items. As such, generalised estimating equations were calculated

regressing coping strategies on the WES subscales and EDSS. Each of these equations were calculated using a Gaussian distribution with an exchangeable working correlation.

## RESULTS

Seventy-nine per cent of the cohort was female, and the median age was 35 years. Respondents had worked in their current ED for a median of 5 years and half worked full time. Average time worked was 0.9 full-time equivalent (FTE) (SD=0.2 FTE). Characteristics of the cohort by study site are provided in table 1. For the nursing cohort, the average age and proportion of females is similar to data available from national surveys in both Australia[30] and Sweden.[31] Similarly, the age and sex of physicians were similar to data reported in Australia,[32] but less information is available from Sweden.

### Content matter experts

The content matter experts were two of the study authors (JC and MCW) who independently reviewed each item in terms of applicability to ED staff. There was initial agreement on 51 (85%) items. Where there was disagreement, this was discussed between the two experts until consensus was achieved. Consensus occurred for all items. This process identified 28 items that were not relevant to the ED context, resulting in a 32-item measure to be subject to factor analyses.

### Exploratory factor analysis

EFA and CFA should not be conducted on the same sample as this results in overfitting and optimistic estimates of model fit within the CFA.[33] As such, the sample was randomly split into equal sized derivation and validation cohorts. EFA was conducted on the derivation cohort to undertake further item reduction and to identify the factor structure of the reduced scale. The number of factors to be retained was identified based on a number of strategies. These included: 1) the number of factors with eigenvalues >1, 2) examination of the scree plot to identify the number of factors where eigenvalues start to level off,[34] 3) parallel analysis, where the number of factors to be retained reflects the factors where eigenvalues are higher than those computed from a simulated random dataset[35] and 4) interpretability of the factor solution. The Kaiser-Meyer-Olkin (KMO) test for sampling adequacy was conducted to determine the suitability of items for factor analysis. Items with a value ≥0.70 show moderate suitability, while values ≥0.80 show good suitability for factor analysis.[36] Values <0.70 were considered for exclusion in the current study. Further item reduction was based on factor loadings. Items that loaded highly (≥0.40) onto one component with low cross-loadings (≤0.20) on other factors were retained.[34] All other items were excluded, unless they were deemed to be an important coping strategy that was not captured by any retained items.

**Table 1** Characteristics of the cohort

| | Entire cohort | ED 1 | ED 2 | ED 3 | ED 4 | ED 5 | ED 6 | ED 7 | ED 8 | ED 9 | ED 10 |
|---|---|---|---|---|---|---|---|---|---|---|---|
| Country | – | Sweden | Sweden | Sweden | Sweden | AUS | AUS | AUS | AUS | AUS | AUS |
| ED type | | Large urban | Tertiary | Small urban | Large urban | Small urban | Large urban | Large urban | Tertiary | Tertiary | Large urban |
| Number of surveys distributed | 1709 | 126 | 145 | 44 | 108 | 196* | 85 | 400 | 150 | 540* | |
| Total responses | 875 | 77 | 92 | 30 | 53 | 44 | 85 | 160 | 93 | 147 | 94 |
| Response rate | 51% | 61% | 63% | 68% | 49% | 66% | | 40% | 62% | 45% | |
| Age, median (IQR) | 35 (28–45) | 44 (32–50) | 36 (28–45) | 42 (33–52) | 36.5 (29–50) | 38 (33–45) | 36 (30–45) | 33 (27–45) | 32 (26–40) | 31 (26–41) | 34 (27–43) |
| Female, n (%) | 653 (79%) | 66 (88%) | 64 (74%) | 29 (97%) | 42 (81%) | 28 (70%) | 56 (74%) | 119 (82%) | 66 (73%) | 108 (77%) | 75 (83%) |
| Nurses, n (%) | 641 (76%) | 70 (97%) | 63 (70%) | 27 (90%) | 40 (78%) | 27 (63%) | 53 (66%) | 114 (75%) | 58 (63%) | 105 (73%) | 84 (91%) |
| Years working in current ED, median (IQR) | 5 (2–10) | 10 (6–20) | 4 (2–8) | 5 (4–8) | 5 (1–13) | 5 (2–11) | 5 (2–10) | 3 (1–7) | 5 (2–10) | 5 (2–10) | 5 (2–10) |
| Years working as a clinician, median (IQR) | 9 (4–16) | 18 (9–32) | 10 (5–17) | 17 (9–30) | 10 (4–34) | 11 (6–13) | 11 (6–13) | 7 (4–12) | 8 (3–13) | 6 (4–14) | 8 (4–16) |
| Number of respondents working full time, n (%) | 371 (47%) | 53 (74%) | 79 (88%) | 20 (71%) | 46 (90%) | 16 (40%) | 29 (36%) | 50 (36%) | 24 (28%) | 37 (31%) | 17 (19%) |

There were missing data for age (n=29), sex (n=49), job role (n=29), years working in ED (n=42), years working as a clinician (n=35),
*Some staff are employed across both sites.
AUS, Australia; ED, Emergency Department; IQR, Interquartile range.

The derivation sample comprised 437 respondents, with a median age of 33 (IQR 27–44), and 78% female. There were 41 (9%) individuals with missing data on any of the JCS items, with the range of missing data for individual items being 1 (0.2% of respondents) to 11 missing responses (3% of respondents). There were no differences in median (or mean) JCS scores for those with and without missing data. As there were minimal missing data on any individual item, and few suitable methods for imputation of missing data in EFA,[37] pairwise deletion of missing data was used for this analysis.

The KMO measure of the 32 JCS items revealed that 30 items were adequate for factor analysis, with the two items focusing on exercising and acceptance having low factorability. As there were remaining items within the JCS that assessed similar coping strategies to these two items, these were removed, leaving 30 items for exploratory factor analysis. Parallel analyses, and scree plots on the 30 items suggested that there were 3–6 factors underlying the items. Factor analyses with oblimin rotation were run using 3, 4, 5 and 6 factors. In each of these analyses, eight items had low loadings on all factors, or had low loadings with high cross-loadings. These items focussed on worry, considering alternative methods for handing the situation, reframing the problem, getting mad, internalising the problem, relaxation, distraction and action planning. Again, these coping strategies were reflected in remaining JCS items and they were removed. The remaining 22 items fell on between 3 and 5 factors. However, the 4-factor and 5-factor solutions displayed factor splitting, with only a small number of items falling with low cross-loadings on the final factor. The 3-factor solution yielded interpretable factors. Based on the 3-factor model, four additional items were removed (discussing the problem with significant others, objective perspective, blaming others and minimising the problem) for low loadings with high cross-loadings. All remaining items were retained as they were felt to be important for the scale. The KMO for these final 18 items was 0.8 while Bartlett's test of sphericity was $\chi^2$=2322, p<0.001. Both indicate adequate factorability. The final solution is provided in table 2. The three factors were labelled problem-focussed coping (4 items), negative emotion-focussed coping (10-items) and positive emotion-focussed coping (4 items).

## Confirmatory factor analysis

This factor structure identified through EFA was validated using confirmatory factor analysis on the remaining cohort. The validation sample comprised 438 individuals with a median age of 36 (IQR=28–46). Within the validation cohort, there were 37 respondents (8%) who had missing data on one or more JCS items. Missing data were imputed using ten datasets that were developed using the remaining JCS items, age, gender and country as predictors. Polychoric correlations were again used and the diagonally weighted least squares estimator was used to estimate model parameters.[38] Robust corrections were used to adjust for any non-independence of errors

**Table 2** Standardised factor loadings for factor structure outlined in exploratory factor analysis

| Factors and items* | Factor loadings | | |
|---|---|---|---|
| | NE | PE | PF |
| Negative emotion-focussed coping | | | |
| 3. Used smoking or medication for stress relief | **0.61** | 0.01 | 0.06 |
| 9. Pessimistic thinking | **0.52** | −0.03 | 0.20 |
| 22. Spent time alone | **0.58** | 0.15 | −0.09 |
| 34. Drank | **0.46** | −0.03 | 0.09 |
| 46. Risky behaviour | **0.61** | 0.04 | 0.05 |
| 48. Ignored problem | **0.58** | 0.11 | −0.25 |
| 51. Self-blame for problem | **0.51** | −0.12 | 0.16 |
| 53. Took stress-reducing medications | **0.61** | −0.10 | 0.20 |
| 56. Physical distancing | **0.80** | −0.06 | −0.13 |
| 58. Wishful thinking | **0.49** | 0.01 | 0.04 |
| Positive emotion-focussed coping | | | |
| 37. Hardiness attitude | 0.14 | **0.48** | 0.07 |
| 39. Used humour | 0.11 | **0.49** | 0.04 |
| 50. Optimistic thinking | −0.05 | **0.82** | 0.05 |
| 54. Refocus on good side | −0.02 | **0.76** | −0.03 |
| Problem-focussed coping | | | |
| 15. Discussed problem with professional | −0.04 | −0.07 | **0.66** |
| 27. Information seeking | 0.07 | 0.13 | **0.63** |
| 42. Discussed problem with someone who has experienced the situation | 0.00 | 0.13 | **0.51** |
| 45. Learnt new skills | −0.04 | 0.26 | **0.43** |

Complete items are from the JCS-A[16] Anne Jalowiec and are available on request. Bolded items are primary factor loading.
*Items have been summarised.
NE, negative emotion-focussed coping; PE, positive emotion-focussed coping; PF, problem-focussed coping.

associated with clustering across sites. However, ICCs were very low for all items (median=2%, IQR=1%–5%), suggesting that there was minimal site-level variance in JCS items.

A variety of goodness-of-fit measures were used in the current study to assess different aspects of model fit. These include absolute fit measures ($\chi^2$ and the standardised root mean square residual (SRMR)), a parsimony correction index (the root mean square error of approximation (RMSEA) and comparative fit indices (the comparative fit index (CFI) and the Tucker-Lewis index (TLI)).[25] Support for the target model is obtained where SRMR is close to 0.08 or below, RMSEA values are

**Table 3** Standardised factor loadings for theorised 3-factor model in confirmatory factor analysis

| | Factor loadings | | |
|---|---|---|---|
| **Factor and items*** | **NE** | **PE** | **PF** |
| Negative emotion-focussed coping | | | |
| 3. Used smoking or medications for stress relief | 0.44 | | |
| 9. Pessimistic thinking | 0.38 | | |
| 22. Spent time alone | 0.63 | | |
| 34. Drank | 0.46 | | |
| 46. Risky behaviour | 0.56 | | |
| 48. Ignored problem | 0.64 | | |
| 51. Self-blame for problem | 0.63 | | |
| 53. Took stress-reducing medications | 0.53 | | |
| 56. Physical distancing | 0.77 | | |
| 58. Wishful thinking | 0.59 | | |
| Positive emotion-focussed coping | | | |
| 37. Hardiness attitude | | 0.55 | |
| 39. Used humour | | 0.56 | |
| 50. Optimistic thinking | | 0.81 | |
| 54. Refocus on good side | | 0.83 | |
| Problem-focussed coping | | | |
| 15. Discussed problem with professional | | | 0.46 |
| 27. Information seeking | | | 0.44 |
| 42. Discussed problem with someone who has experienced the situation | | | 0.59 |
| 45. Learnt new skills | | | 0.57 |

Complete items are from the JCS-A[16] Anne Jalowiec and are available on request.
*Items have been summarised.
NE, negative emotion-focussed coping; PE, positive emotion-focussed coping; PF, problem-focussed coping.

close to 0.06 or below and CFI and TLI values are close to 0.90 or greater.[39] Fitting the three-factor solution to the validation sample provided a good fit to the data ($\chi^2$=323, p<0.01, CFI=0.93, TLI=0.92, SRMR=0.08, RMSEA=0.06) and all items loaded significantly onto their respective factors. Standardised loadings are provided in table 3.

To further explore the data, a number of additional (post hoc) models were fit to identify whether the factor structure was similar across job roles and across country.

Given sample size limitations for some groups, these analyses were fit on the entire cohort rather than focusing only on the validation cohort. For each comparison, several invariance models were fit in the order specified by Wu and Estabrook.[40] Goodness-of-fit measures were compared across models requiring increasing invariance, including invariance of factor structure, thresholds and loadings. For job role, the goodness-of-fit measures for all models were similar, indicating that both factor structure and factor loadings were similar for doctors and nurses (online supplementary table 1). For country, goodness-of-fit measures were acceptable for all models. However, they were best for the factor structure model. This indicates that the factor structure was similar for Swedish and Australian respondents, but the strength of individual factor loadings were slightly different across countries. Factor loadings for models where loadings were allowed to vary are provided in online supplementary tables 2 and 3) and show only minor differences in factor loadings across groups.

### Descriptive data and criterion validity

Descriptive data for the overall cohort and by country are reported in table 4. Cronbach's alphas were 0.77 for negative-emotion focussed coping, 0.68 for positive-emotion focussed coping and 0.61 for problem-focussed coping. Alphas could not be improved through the removal of any individual item. Negative emotion-focussed coping strategies were used less often than positive emotion-focussed coping or problem-focussed coping strategies. The cohort from Sweden reported slightly less use of negative coping strategies and slightly more use of positive and problem-focussed strategies.

Coefficients from the regression of coping strategies on WES-10 subscales are provided in table 5. There was an association between the use of negative emotion-focussed coping strategies and poor perceptions of the work environment (lower opportunity for personal and professional growth, higher workload, higher conflict among staff members and higher nervousness or tension about going to work). The use of either problem-focussed coping strategies or positive emotion-focussed coping strategies had a weak association with perceptions that the workplace provided opportunity for personal and professional growth (self-realisation).

Coefficients from the regression of coping strategies on ratings of stressors are provided in table 5. Respondents

**Table 4** Coping strategies by country

| | Overall cohort | Australia | Sweden | Difference (95% CI of difference) |
|---|---|---|---|---|
| Negative emotion-focussed coping | 1.0 (0.5) | 1.1 (0.5) | 0.8 (0.4) | −0.3 (−0.3 to −0.2) |
| Positive emotion-focussed coping | 2.1 (0.6) | 2.1 (0.5) | 2.3 (0.6) | 0.2 (0.1 to 0.3) |
| Problem-focussed coping | 1.9 (0.6) | 1.9 (0.6) | 2.1 (0.5) | 0.2 (0.1 to 0.3) |

The scale ranges from 0 (never used) to 3 (often used).

**Table 5** Regression coefficients and 95% CIs from coping strategies with WES-10 subscales and job stressors

| | Negative emotion-focussed coping | Positive emotion-Focussed coping | Problem-focussed coping |
|---|---|---|---|
| **WES-10 subscales** | | | |
| Self-realisation | −0.3 (-0.4 to -0.2) | 0.2 (0.1 to 0.2) | 0.1 (0.1 to 0.2) |
| Workload | 0.3 (0.2 to 0.3) | 0 (0 to 0.1) | 0.1 (0 to 0.2) |
| Conflict | 0.5 (0.4 to 0.6) | 0 (−0.1 to 0.1) | 0.1 (0 to 0.2) |
| Nervousness | 0.6 (0.5 to 0.7) | −0.1 (−0.2 to 0) | 0 (−0.1 to 0.1) |
| **EDSS** | | | |
| Workplace violence | 0.9 (0.4 to 1.4) | 0.2 (−0.3 to 0.6) | 0.2 (−0.2 to 0.7) |
| Heavy workload | 1.5 (1.1 to 1.8) | −0.2 (−0.6 to 0.1) | 0.2 (−0.2 to 0.5) |
| High acuity patients | 1.4 (1.0 to 1.8) | −0.3 (−0.7 to 0.1) | 0 (−0.4 to 0.4) |
| Inability to provide optimum care | 1.5 (1.1 to 1.9) | 0 (−0.3 to 0.4) | 0.7 (0.3 to 1.1) |
| Environmental concerns | 1.6 (1.1 to 2.0) | −0.1 (−0.5 to 0.3) | 0.4 (0.0 to 0.8) |

The coefficient represents the predicted change in WES-10 scores or ED stressor scales for each 1-point increase in coping strategies.
EDSS, emergency department stressor scale; WES, work environment scale (10 items).

reporting higher use of negative emotion-focussed coping strategies rated events to be more stressful. There was no association between ratings of stressors and either positive emotion-focussed coping or problem-focussed coping. One exception was that higher problem-focussed coping had a weak relationship with higher stress over inability to provide optimum care.

## DISCUSSION

This study used data from a large sample of international ED staff to develop a refined JCS for future use by ED staff. The final scale incorporated 18 items measuring three styles of coping: problem-focussed coping, positive emotion-focussed coping and negative emotion-focussed coping. It is short enough to encourage inclusion in future surveys exploring ED staff perceptions. Data exploration using this tool indicated that the use of negative coping strategies was associated with poor perceptions of the work environment and higher ratings of ED job stressors in both Australian and Swedish EDs. This modified scale is likely to be useful in identifying the types of coping strategies employed by ED staff and for use in evaluating interventions to reduce the use of maladaptive coping strategies.

The three categories of coping identified in the present study: problem-focussed coping, positive emotion-focussed coping and negative emotion-focussed coping aligned with other theoretical literature. For example, the division of problem-focussed and emotion-focussed coping accords with Lazarus's theoretical model of coping.[5] This categorisation also acknowledges that there are different styles of emotion-focussed coping, some potentially adaptive and others potentially maladaptive.[8] By developing a shortened JCS that corresponds with theoretical and empirical research on coping, we have developed an ED-based tool to explore coping strategies,

and a tool that is likely to be suitable for administration more broadly in the busy environments experienced by acute care health professionals. However, further validation would be required to identify broader utility of this scale.

Using the modified JCS enabled the team to establish an association between negative emotion-focussed coping and poor perceptions of the working environment, as well as higher ratings of stress around listed events. This pattern of response is consistent with previous research[3 9 41] and accords with pragmatic perceptions of the development of burnout in ED staff. Maladaptive coping strategies have been consistently linked to poor perceptions of stress management,[41] poor psychological health[3] and burnout.[9] In contrast, problem-focussed and positive emotion-focussed strategies were largely unrelated to perceptions of the working environment and perceptions of stress around specific events. This finding may appear to differ from previous studies finding that problem-focussed strategies in particular have beneficial outcomes, such as lower burnout and distress.[3 9] However, this study focussed on ratings of stressors, rather than the outcomes of stress (eg, psychological distress and burnout). Additional research relating these scales to health outcomes would be required to clarify this relationship. Furthermore, the broader literature does note that problem-focussed and emotion-focussed strategies may not be classed as inherently good or bad.[5] Instead, their effectiveness depends on the situation.[8] In situations where stressors are controllable (eg, taking an exam), problem-focussed coping is useful.[42] Conversely, when dealing with stressors such as a major loss, it may be adaptive initially to engage in some palliative coping.[43] As such, these strategies may not show clear relationships with broad outcome variables.

The strengths of this study include the generation of a validated revised JCS using a large multidisciplinary international sample. The limitations include the lack of availability of longitudinal data, meaning that test–retest reliability was unable to be determined. Test–retest reliability is necessary to ensure that the scale is reliable and stable in assessing coping strategies across time. The response rate was reasonable for a survey design. However, it is unknown whether the respondents were representative of the broader population. No data were available on the specific response rate for nurses and physicians. Nurses made up the majority of the sample at a number of sites and it is unclear whether this accurately reflects the workforce, or whether there was low response rate by physicians in those sites. No data were available on burnout or employee mental health and so limited assessment of the outcomes of coping could be conducted. The JCS assesses coping using a 4-item Likert scale. Research has shown that the reliability, validity and discriminating power of 4-point scale is relatively poor, with the optimum number of categories being between 7 and 10.[44] Future research may benefit from incorporating a larger number of response categories.

The ED can be a particularly stressful workplace. Stressors can result in increased sick leave and staff burnout, poor staff recruitment and retention, decreased staff morale and decreased job satisfaction. Understanding ways in which ED staff cope with stressors that pertain to their work is important. We have developed a revised JCS to assess the coping strategies used by ED staff. This shortened version of the JCS incorporates ten maladaptive emotion-focussed strategies incorporating aspects such as risky behaviour, drinking, stress-reducing medications and ignoring the problem. It also includes four positive emotion-focussed strategies around refocusing, being optimistic and using humour. Finally, the scale includes four problem-focussed coping strategies including information seeking and learning new skills. This scale may be used to evaluate the outcome of interventions that seek to promote positive coping strategies.

**Author affiliations**
[1]Institute of Health and Biomedical Innovation, School of Public Health annd Social Work, Queensland University of Technology Faculty of Health, Kelvin Grove, Queensland, Australia
[2]Emergency and Trauma Centre, Royal Brisbane and Women's Hospital, Herston, Queensland, Australia
[3]School of Nursing & Midwifery, University of the Sunshine Coast, Maroochydore, Queensland, Australia
[4]Department of Emergency Medicine, Princess Alexandra Hospital, Woolloongabba, Queensland, Australia
[5]School of Nursing, Midwifery, and Social Work, University of Queensland—St Lucia Campus, Brisbane, Queensland, Australia
[6]Health and Crisis Management and Policy, Sahlgrenska Akademin, Goteborgs Universitet, Goteborg, Sweden
[7]Department of Health and Medical Sciences, Linköping University, Linkoping, Sweden
[8]Department of Emergency Medicine, Local Health Care Services in Central Östergötland, Linkoping, Sweden
[9]Department of Emergency Medicine, Sunshine Coast Hospital and Health Service, Nambour, Queensland, Australia
[10]Menzies Health Institute, Griffith University, Nathan, Queensland, Australia
[11]Department of Emergency Medicine, Gold Coast Health Service District, Southport, Queensland, Australia

**Collaborators** The WES Investigators: Hui (Grace) Xu, Elizabeth Elder, James Hughes, Monica A Magnusson.

**Contributors** JG, MCW, AJ, EC, DW, OT, LA and JC conceived and designed the study. MCW, AJ, EC, DW, OT, LA and JC were involved in acquisition of data. JG analysed the data. MCW, AJ, EC, DW and JC assisted with interpretation of the data. JG, MCW and JC drafted the work and all authors revised the manuscript critically for important intellectual content.

**Funding** This study was supported by an Emergency Medicine Foundation grant (EMSS-410R22-2014).

**Competing interests** None declared.

**Patient consent for publication** Not required.

**Ethics approval** This study was carried out in accordance with the code of ethics of the world medical association (Declaration of Helsinki). Hospital and University human research and ethics committees (HREC/14/QGC/173; NRS/15/16/HREC) approved the protocol for the Australian sites and the Regional Ethical Review Board in Linköping (permit number 2018/563–32) approved the protocol for the Swedish sites. Informed consent was implied through the completion and return of the survey.

**Provenance and peer review** Not commissioned; externally peer reviewed.

**Data availability statement** No data are available.

**ORCID iD**
Jaimi H Greenslade http://orcid.org/0000-0002-6970-5573

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
