## [Reviewer comments · BMJ Open]

ARTICLE DETAILS

TITLE (PROVISIONAL)	Development of a revised Jalowiec coping scale for use by emergency clinicians: A cross-sectional scale development study
AUTHORS	Greenslade, Jaimi; Wallis, Marianne C; Johnston, Amy; Carlström, Eric; Wilhelms, Daniel; Thom, Ogilvie; Abraham, Louisa; Crilly, Julia

VERSION 1 – REVIEW

REVIEWER	Marianna Masiero Department of Biomedical and Clinical Sciences, University of Milan, Italy
REVIEW RETURNED	09-Aug-2019

GENERAL COMMENTS	Thank you for the opportunity to revise the manuscript (ID BMJOPEN-2019-033053) title “Development of a revised Jalowiec coping scale for use by emergency clinicians: A scale development study”. The manuscript discusses an important issue in health domain: coping strategies in Emergency Departments (EDs). However, the manuscript contains several limitations that required pivotal changes. Abstract Abstract is poor and it does not provide sufficient theoretical and methodological information to help the readers to understand the outcomes and the study design. For example, Authors did not provide any statistical information to understand the characteristics of the sample (mean age, the composition for gender etc.). The same in the setting paragraph only information available concerned nationality (Australian and Sweden). In addition, I read in the main document that other questionnaires are used. Notwithstanding, in the abstract, there is no information about them. Finally, I think that the last sentence exceeded the scope of the current study “This will aid in the development and evaluation of evidence-based interventions to promote positive coping while reducing the use of maladaptive coping strategies” (Page 3 – Line [53-56]). Introduction Overall, I think that a solid theoretical background on coping strategy and coping strategy applied to the health domain should be added. [ ] More sentences are generic and no supported by adequate references. For example, Authors reported that accruing evidence observed an association between problem-focused coping and lower level of burnout (Page 5 – Line [35-36]), but they did not explain in which way this happens.
---

	[ ] Authors wrote that “existing scales are lengthy, or have displayed poor psychometric properties” (Page 5 Line [48-49]). I think that this sentence should be explained better: which kind of psychometric properties? Validity’s problem? Problem-related to the sample characteristics? Limitations due to the absence of test re-test? I think that should be explained in-depth literature gap reported in order to understand better the strengthens of this study. [ ] Finally, another concern is related to the main aims of this study. In my opinion, the primary and secondary aims are not well explained. Firstly, Authors affirmed that they want to “develop” a scale for assessing coping strategies in ED staff; and then they said that they want to use this modified JCS-ED to describe the coping strategies in ED staff in two different countries: Australia and Sweden. The aims should be re-formulated. Method [ ] Authors should provide clear information about the procedure used (e.g., “How has been scale submitted to participants?”); [ ] Concerning participants: why the study was conducted on two different population Australian and Sweden? [ ] No information is provided about the inclusion and exclusion criteria (Specialization? Experience in EDs?). For example, a long experience in ED may affect the coping strategy of the doctors and nurses. [ ] The instruments (WES-10, ED stressor scale) used should be described better: are validated for Australian and Sweden population? Which are the Cronbach’s value for each scale? [ ] Data analysis paragraph is too long many information should be replaced in the results section. In my opinion, data analysis paragraph should be contained information about the statistical plan that researchers want to apply. Also, it should be written in brief, but clear way. Results [ ] More information should be provided about item reduction and consensus by content matter experts (e.g., level of agreement between the experts, how disagreements were resolved, etc.) [ ] No information is provided about the difference between doctors and nurses, as well as, between male and female. [ ] Cronbach's alpha should be analyzed and reported. [ ] In order to increase understanding items of the new version of the scale should be provided. Conclusions Lastly, Authors should re-write the conclusions describing in depth the new scale obtained.
--	---

REVIEWER	Matt Grace Hamilton College, USA
REVIEW RETURNED	19-Aug-2019

GENERAL COMMENTS	Summary: Thank you for the opportunity to review “Development of a revised Jalowiec coping scale for use by emergency clinicians: A scale development study.” The authors draw upon cross-sectional data collected from a cross-national sample (from Australia and Sweden) to validate a shorter, more practical version of the JCS-ED. Employing exploratory data analysis, the authors find support for a 3-factor, 18-item version of the scale. Subsequent confirmatory analyses substantiated the use of the items included
---

in the revised scale. While the study offers important insights for applied researchers and academic medicine more broadly, I have a number of comments and questions for the authors to consider.

Methods:

1. The authors report the overall response rate and the response rate at each research site. I commend the authors on the study's high response rate but wondered how the overall response rate and site-specific response rates for doctors and nurses compared. Given that doctors are notoriously difficult to reach in studies of this nature, and that nurses made up the vast majority of the sample at certain sites (e.g., 97% at ED1, 90% at ED3, 91% at ED10) this information would be helpful to know as a reader.

2. Did participants receive any compensation for their participation? If so, this information should be included.

Exploratory Factor Analysis:

1. The authors provide a detailed overview of their analysis and rigorously employ multiple methods (eigenvalues, screeplots, etc.) to determine which factors to retain. Given that nurses and doctors are markedly different in terms of the content of their work, level of autonomy, and stress exposure, I wondered how factor loadings compared when factor analyses were conducted using separate occupational subsamples (i.e., only doctors v. only nurses)? Similarly, given the differences in social policy and healthcare administration across countries, how do factor loadings compare across country samples?

Descriptive Data and Criterion Validity:

1. In regards to predictive validity, the authors regressed the JCS on the WES-10 and EDSS (it would be helpful if p-values were included in the text while describing these relationships). Given the connections drawn between coping styles and burnout/psychological distress, I wondered if the authors had available data to examine whether the reduced scale was correlated with any form of mental health measure. If not, I would minimize the connections drawn to mental health throughout the paper as the authors cannot comment on potential associations to these outcomes.

Discussion:

1. The claim that the lack of association between problem-focused / positive emotion-focused strategies and perceptions of stress provides evidence contravening past studies linking these factors to lower burnout and reduced distress is overstated given that the authors did not measure either of these constructs. The literature on stress theory (see Cohen and McKay, Pearlin) distinguishes between stressors (social threats that prompt emotional/psychological adaption) and psychological distress.

2. In terms of the assertion that it is unknown whether respondents are representative of the broader population, this is certainly the case as the authors are not employing a random sample of emergency staff in these countries. Still, are there statistics on the demographics of emergency room nurses and doctors in this nation that would provide some indication of the extent to which the demographics of the analytic sample aligns with workers in each respective country?

REVIEWER	Dr. Razieh Froutan Razieh froutan Ph.D Assistant Professor in Nursing Management School of Nursing and Midwifery Mashhad University of Medical Sciences Ebne- Sina Street Mashhad, Iran Tel: 0098 511 8539775 Fax: 0098 511 8539775
REVIEW RETURNED	12-Sep-2019

GENERAL COMMENTS	Dear Bern Thank you for your email. The reliability of the scale should have been examined by measuring its internal consistency and stability. Cronbach's alpha and theta coefficients should have been used to examine the internal consistency of the scale. The stability of the scale should have been examined using the test-retest method, and intraclass correlation coefficient (ICC) should have been used to measure its stability. KMO test and Bartlett sphericity have not been reported. The correlation of the two scale(60-items with 18 items) should be examined.
--

VERSION 1 – AUTHOR RESPONSE

Reviewer: 1

Thank you for the opportunity to revise the manuscript (ID BMJOPEN-2019-033053) title “Development of a revised Jalowiec coping scale for use by emergency clinicians: A scale development study”. The manuscript discusses an important issue in health domain: coping strategies in Emergency Departments (EDs). However, the manuscript contains several limitations that required pivotal changes.

Abstract

Abstract is poor and it does not provide sufficient theoretical and methodological information to help the readers to understand the outcomes and the study design. For example, Authors did not provide any statistical information to understand the characteristics of the sample (mean age, the composition for gender etc.). The same in the setting paragraph only information available concerned nationality (Australian and Sweden). In addition, I read in the main document that other questionnaires are used. Notwithstanding, in the abstract, there is no information about them. Finally, I think that the last sentence exceeded the scope of the current study “This will aid in the development and evaluation of evidence-based interventions to promote positive coping while reducing the use of maladaptive coping strategies” (Page 3 – Line [53-56]).

We have now made several changes to the abstract (page 3). Please note that with these changes, the abstract substantially exceeds the 300-word limit allowed by BMJ open. We hope that this is acceptable to the editor. Specific changes are as follows:

We have now included statistical information about the characteristics of the sample. Specifically, in the participants section, we have stated:

“The median age of participants was 35 years (IQR: 28-45 years) and they had been working in the ED for a median of 5 years (IQR: 2-10 years). Seventy-nine percent were female and 76% were nurses”

In the design section, we have stated:

“A cross-sectional survey incorporating the Jalowiec coping scale, the working environment scale-10 (WES-10) scale and a measure of workplace stressors was administered.”

In the settings, we have now stated:

“There were three tertiary hospitals, five large-urban hospitals and two small urban hospitals”

We have also removed the last sentence of the abstract about the development and evaluation of interventions.

Introduction

Overall, I think that a solid theoretical background on coping strategy and coping strategy applied to the health domain should be added.

–More sentences are generic and no supported by adequate references. For example, Authors reported that accruing evidence observed an association between problem-focused coping and lower level of burnout (Page 5 – Line [35-36]), but they did not explain in which way this happens.

We have expanded the discussion around the transactional model of stress to explain this further. This section (page 5, paragraph 2) now reads as follows:

“The transactional model of stress and coping outlines the processes by which stressful situations give rise to coping behaviours, and ultimately to workplace well-being outcomes [5]. Specifically, this model posits that individuals appraise or evaluate their situation. If a situation is deemed taxing or overwhelming, they will engage in thoughts or actions (coping strategies) to manage that situation [6,7]. Many different coping strategies can be employed, and these depend on both the environment and on personal disposition [8]. Such coping strategies serve to alter stress in various ways. First, they may address the problem causing distress (problem-focussed coping) [8]. Problem-focussed coping attempts to removes the source of distress, thereby removing the stressor [6]. As such,

individuals who successfully utilise problem-focussed coping experience lower distress and, thus, experience fewer negative outcomes from stress [6]. Second, coping strategies may attempt to ameliorate the negative emotions associated with the stressor (emotion-focussed coping) [8]. Emotion focussed coping can change the way individuals think about or interpret what is happening [6]. Some emotion focussed coping strategies may be positive and functional, while others can have negative consequences [8]. For example, drinking to cope may provide short term relief from stress, but does not actually reduce the problem in the longer term. Such maladaptive coping does not reduce the impact of the stressful situation and, thus, is linked to poorer outcomes [8]. In line with this theory, previous studies have suggested that ED staff using problem-focussed coping have lower levels of burnout [9] and better psychological health [3] than those using emotion-oriented coping or maladaptive coping strategies.”

– Authors wrote that “existing scales are lengthy, or have displayed poor psychometric properties” (Page 5 Line [48-49]). I think that this sentence should be explained better: which kind of psychometric properties? Validity’s problem? Problem-related to the sample characteristics? Limitations due to the absence of test re-test? I think that should be explained in-depth literature gap reported in order to understand better the strengthens of this study.

We have now expanded this paragraph to note that there are no existing ED scales. We have also reported additional details regarding psychometric properties of the most commonly used coping scale, and about the Jalowiec scale. Specifically, we have stated on page 6, first paragraph:

There are no validated scales of which we are aware that were designed to measure coping within the ED. As such, various generic coping scales have been applied to this setting [e.g. 9,10, 11, 12]. Existing scales are lengthy [8] or have displayed poor psychometric properties [13]. For example, the COPE questionnaire is the most commonly used instrument in adult samples [14]. This questionnaire is a 53-item index, but has been found to have an equivocal factor structure with moderate reliability [14,15]. The Jalowiec Coping Scale (JCS) is another coping scale, with the advantage that it was specifically developed to measure the process of coping in a healthcare setting. This scale incorporates 60 items distributed across 8 dimensions and has been translated into more than 20 languages [16]. The JCS was originally developed to measure problem- and emotion-focused coping styles, but the final version identifies 8 different dimensions of coping. The JCS has been used nationally and internationally with both patients and employees in the healthcare setting [17]. While this scale has been extensively utilized, studies assessing its psychometric properties have not supported the 8-item structure [18]. Further, a 60-item scale is arguably too long for busy ED staff.

– Finally, another concern is related to the main aims of this study. In my opinion, the primary and secondary aims are not well explained. Firstly, Authors affirmed that they want to “develop” a scale for assessing coping strategies in ED staff; and then they said that they want to use this modified JCS-ED to describe the coping strategies in ED staff in two different countries: Australia and Sweden. The aims should be re-formulated.

We have now reworded the aims (page 6, second paragraph) as follows:

The overarching goal of this study was to produce a coping scale that would be useful for assessing

coping in ED staff. This instrument should incorporate items to identify beneficial and maladaptive problem- and emotion-focused coping strategies while also being short enough for easy completion in a busy work environment. To achieve this goal, we sought to develop and validate a modified JCS coping scale for the emergency department setting.

Method

- Authors should provide clear information about the procedure used (e.g., “How has been scale submitted to participants?”);

We have modified the participants and design section (page 6, last paragraph) to clarify that the paper surveys were hand distributed to participants by a local investigator at each site. Surveys were returned using locked boxes or using stamped self-addressed envelopes. This section now reads as follows:

A cross-sectional paper survey was administered between July 2016 and June 2017 depending on logistics at each site. Surveys were hand distributed to 1709 staff by a local investigator at each site, with 876 returned. One survey was excluded due to extensive missing data, making the final sample size 875 (51% response rate). Staff were also provided with information and invited to participate via email and in ward-based information sessions. Surveys were returned to locked boxes within each hospital ED, or via stamped self-addressed envelopes. A reminder email was sent out two-weeks after survey distribution.

- Concerning participants: why the study was conducted on two different population Australian and Sweden?

Using samples from different healthcare systems representing different traditions and cultures was regarded as a factor that would improve the generalizability of this study. The decision to use Australia and Sweden specifically was pragmatic. The primary and senior authors both worked in Australian Emergency Departments. The senior author has developed collaborative relationships with several investigators in Sweden. These collaborations enabled the survey to be administered in two countries.

- No information is provided about the inclusion and exclusion criteria (Specialization? Experience in EDs?). For example, a long experience in ED may affect the coping strategy of the doctors and nurses.

There were no exclusion criteria. All full-time and part-time medical and nursing staff employed within the study EDs were eligible for inclusion in this study. We wished to include as broad a sample as possible to identify the coping strategies used by the complete range of staff in Emergency. We have now clarified this in the participants section (page 6, last paragraph) by stating “there were no exclusion criteria”

- The instruments (WES-10, ED stressor scale) used should be described better: are validated for Australian and Sweden population? Which are the Cronbach’s value for each scale?

We have now included the following information in the instrument section (page 7, 3rd paragraph) about the WES-10:

The WES-10 is a 10-item scale that describes four aspects of the working environment; opportunity for personal and professional growth (self-realization, 4 items); workload (2 items); interpersonal conflict (2 items); and nervousness (2 items) [19]. Respondents are asked to answer how they feel about each item on 1 to 5 scale, with Likert-scale labels differing according to the question asked. This scale has been used within an ED [20] but was not specifically developed or validated in this setting. Cronbach alpha for self-realization was 0.72 while Spearman-Brown was 0.64 for workload, 0.57 for conflict and 0.70 for nervousness. These show moderate internal consistency and are in line with reliability coefficients reported in previous studies [19,20].

For the ED stressor scale, Cronbach alpha is not reported as items are not combined into subscales. We have now provided further information on this scale as follows:

“Job stressors were assessed using a 15-item ED stressor scale (EDSS) [21]. This scale was designed to assess stressors reported by nurses within an Australian ED [21]. Respondents were asked to rate on a scale of 1-15, how stressful they would find each of 15 stress-provoking events. They were also asked how often they experienced each event, ranging from 0 (never) to 3 (daily). Items on this scale are not combined to form sub-scales, they each assess a different stressor within the ED.”

– Data analysis paragraph is too long many information should be replaced in the results section. In my opinion, data analysis paragraph should be contained information about the statistical plan that researchers want to apply. Also, it should be written in brief, but clear way.

The data analysis paragraph has been shortened considerably. Information on missing data and details about factor analyses have been moved to the results.

Results

– More information should be provided about item reduction and consensus by content matter experts (e.g., level of agreement between the experts, how disagreements were resolved, etc.)

In the section on content matter experts (page 11, first paragraph), we have noted

“There was initial agreement on 51 (85%) items. Where there was disagreement, this was discussed between the two experts until consensus was achieved. Consensus occurred for all items”.

- No information is provided about the difference between doctors and nurses, as well as, between male and female.

As per the comments made by reviewer 2, we have now included supplementary analyses examining differences in factor analyses between doctors and nurses and by country. Further details of these analyses are provided below. We found that the factor analysis for doctors and nurses was similar. With regard to sex, there is a large overlap between sex and job role. As such, we have not reported sex differences separately.

- Cronbach's alpha should be analyzed and reported.

We have now included the following in the description data and criterion validity section (page 15, final paragraph):

“Cronbach alphas were 0.77 for negative-emotion focussed coping, 0.68 for positive-emotion focussed coping, and 0.61 for problem-focussed coping. Alphas could not be improved through the removal of any individual item”

- In order to increase understanding items of the new version of the scale should be provided.

We are unable to provide the complete items for publication as these remain copyrighted by Anne Jalowiec. Full items are available on request and this has now been noted under each table.

Conclusions

Lastly, Authors should re-write the conclusions describing in depth the new scale obtained.

We have now included the following information in the conclusion (page 19, final paragraph):

“This shortened version of the JCS incorporates ten maladaptive emotion focussed strategies incorporating aspects such as risky behavior, drinking, stress-reducing medications and ignoring the problem. It also includes four positive emotion focussed strategies around refocusing, being optimistic and using humour. Finally, the scale includes four problem-focussed coping strategies including information seeking and learning new skills”

Reviewer: 2

Methods:

- 1) The authors report the overall response rate and the response rate at each research site. I commend the authors on the study's high response rate but wondered how the overall response rate and site-specific response rates for doctors and nurses compared. Given that doctors are notoriously difficult to reach in studies of this nature, and that nurses made up the vast majority of the sample at certain sites (e.g., 97% at ED1, 90% at ED3, 91% at ED10) this information would be helpful to know as a reader.

Unfortunately, there are no data on the overall response rate for doctors and nurses. Anecdotally, we believe that several of the Swedish sites (where the cohort was predominantly nurses) had lower numbers of permanent ED medical staff, with physicians from other specialties also temporarily contributing to the ED workforce. Unfortunately, we were unable to obtain accurate information on how many physicians and nurses were employed in each site to support this belief. We have now included a limitation (page 19, first paragraph) as follows:

"No data were available on the specific response rate for nurses and physicians. Nurses made up the majority of the sample at a number of sites and it is unclear whether this accurately reflects the workforce, or whether there was low response rate by physicians in those sites"

- 1) Did participants receive any compensation for their participation? If so, this information should be included.

There was no compensation for participation in this study. We have now included in the participants and design section (page 7 first paragraph). *"No compensation was provided to staff for completing the survey"*

Exploratory Factor Analysis:

- 3) The authors provide a detailed overview of their analysis and rigorously employ multiple methods (eigenvalues, screeplots, etc.) to determine which factors to retain. Given that nurses and doctors are markedly different in terms of the content of their work, level of autonomy, and stress exposure, I wondered how factor loadings compared when factor analyses were conducted using separate occupational subsamples (i.e., only doctors v. only nurses)? Similarly, given the differences in social policy and healthcare administration across countries, how do factor loadings compare across country samples?

We agree that there may be differences across subgroups. Unfortunately, we do not have adequate numbers in either the derivation or validation cohorts to run separate factor analyses, particularly for doctors. However, as part of the confirmatory factor analyses, we have now included a number of supplementary analyses focussing on differences by job and country. For these analyses we used the entire cohort (not broken down into derivation and validation cohorts). We have done this analysis as a CFA rather than an EFA as CFA has well established procedures for comparing subgroups. The information in text (page 15, second paragraph) is as follows:

“To further explore the data, a number of additional (post hoc) models were fit to identify whether the factor structure was similar across job roles and across country. Given sample size limitations for some groups, these analyses were fit on the entire cohort rather than focusing only on the validation cohort. For each comparison, several invariance models were fit in the order specified by Wu and Estabrook [40]. Goodness of fit measures were compared across models requiring increasing invariance, including invariance of factor structure, thresholds, and loadings. For job role, the goodness of fit measures for all models were similar, indicating that both factor structure and factor loadings were similar for doctors and nurses (Supplementary Table 1). For country, goodness of fit measures were acceptable for all models. However, they were best for the factor structure model. This indicates that the factor structure was similar for Swedish and Australian respondents, but the strength of individual factor loadings were slightly different across countries. Factor loadings for models where loadings were allowed to vary are provided in Supplementary Tables 2 and 3) and show only minor differences in factor loadings across groups”

Descriptive Data and Criterion Validity:

- 4) In regards to predictive validity, the authors regressed the JCS on the WES-10 and EDSS (it would be helpful if p-values were included in the text while describing these relationships). Given the connections drawn between coping styles and burnout/psychological distress, I wondered if the authors had available data to examine whether the reduced scale was correlated with any form of mental health measure. If not, I would minimize the connections drawn to mental health throughout the paper as the authors cannot comment on potential associations to these outcomes.

We would prefer not to include p values in the text. This study has a large sample size when considering individual predictors of one outcome. As such, all relationships discussed in text have a $p < 0.01$. Examination of the regression coefficients shows that some of the relationships are quite weak, and we did not want these over-interpreted by a strong p value. However, we will reconsider including these if the reviewer feels strongly about this point.

We do not have any data available on a mental health measure. We have now changed several aspects of the discussion to reflect this. Specifically, as outlined in response to your next point, we have now changed the discussion around the relationship between coping strategies and stressors. We also changed a sentence in the discussion that referred to the mental health outcomes of emotion-focussed coping. It now states that *“there are different styles of emotion-focussed coping, some potentially adaptive and others potentially maladaptive”*. We also have noted this as a limitation *“No data were available on burnout or employee mental health and so limited assessment of the outcomes of coping could be conducted.”*

Discussion:

- 5) The claim that the lack of association between problem-focused / positive emotion-focused strategies and perceptions of stress provides evidence contravening past studies linking these factors to lower burnout and reduced distress is overstated given that the authors did not measure either of these constructs. The literature on stress theory (see Cohen and McKay, Pearlin) distinguishes between stressors (social threats that prompt emotional/psychological adaption) and psychological distress.

Thank you for this point. We have now changed this section of the discussion (page 18, second paragraph) to state *“This finding may appear to differ from previous studies finding that problem-focused strategies in particular have beneficial outcomes, such as lower burnout and distress [3,9]. However, this study focussed on ratings of stressors, rather than the outcomes of stress (e.g., psychological distress and burnout). Additional research relating these scales to health outcomes would be required to clarify this relationship”*.

60 In terms of the assertion that it is unknown whether respondents are representative of the broader population, this is certainly the case as the authors are not employing a random sample of emergency staff in these countries. Still, are there statistics on the demographics of emergency room nurses and doctors in this nation that would provide some indication of the extent to which the demographics of the analytic sample aligns with workers in each respective country?

Unfortunately, there is no comprehensive source of information on demographics of emergency staff in the two countries. However, we have the following comparisons:

Australia

- The 2015 Australian Institute of Health and Welfare (AIHW) survey of nurses found that the average age of emergency nurses was 39 years. The average age of nurses in our cohort was 36 years.
- In 2015 AIHW survey of nurses found that 90% of all nurses were female (no specific breakdown was provided for emergency staff). 88% of our nursing cohort were female.
- The 2015 AIHW survey of medical practitioners reported that emergency specialists were an average of 45 years old and 32% were female. The Australian specialists in our cohort were an average age of 44 years and 29% were female.
- The 2015 AIHW survey of medical practitioners reported that specialists in training were an average of 34 years and 51% were female. In our cohort, the average age of Australian specialists in training was 33 years and 53% were female.

Sweden:

- In 2016, 48% of all physicians (no breakdown provided for emergency staff) were female. In our cohort, 57% of Swedish emergency physicians were female.
- A published paper survey of Swedish nurses (no breakdown for emergency staff) found the average age was 40 years and 94% were female. The average age in our cohort of Swedish nurses was 42 years with 88% female.

Within the results section (page 9, second paragraph), we have now stated:

“For the nursing cohort, the average age and proportion of females is similar to data available from national surveys in both Australia [30] and Sweden [31] Similarly, the age and sex of physicians were similar to data reported in Australia [32], but less information is available from Sweden”.

Reviewer: 3

The reliability of the scale should have been examined by measuring its internal consistency and stability. Cronbach's alpha and theta coefficients should have been used to examine the internal consistency of the scale. The stability of the scale should have been examined using the test-retest method, and intraclass correlation coefficient (ICC) should have been used to measure its stability.

KMO test and Bartlett sphericity have not been reported. The correlation of the two scale(60-items with 18 items) should be examined.

Cronbach alphas have now been included in the descriptive data and criterion validity section (as outlined in the response to reviewer 1). The survey was only administered to participants once. As such, we could not measure the stability of the survey using the test-retest measure. This has been noted in the limitations section (page 18, final paragraph) as follows

“The limitations include the lack of availability of longitudinal data, meaning that test-retest reliability was unable to be determined. Test-retest reliability is necessary to ensure that the scale is reliable and stable in assessing coping strategies across time”

The KMO and Bartlett's test have now been reported (page 12, final paragraph). Specifically, in the exploratory factor analysis section of the results, we have now stated: *“The KMO for these final 18 items was 0.8 while Bartlett's test of sphericity was 2322, $p < 0.001$. Both indicate adequate factorability”*

VERSION 2 – REVIEW

REVIEWER	Matt Grace Hamilton College, USA
REVIEW RETURNED	09-Oct-2019

GENERAL COMMENTS	Thank you for the opportunity to review “Development of a revised Jalowiec coping scale for use by emergency clinicians: A cross-sectional scale development study.” The authors draw upon cross-sectional data collected from a cross-national sample (Australia and Sweden) to validate a shorter, more logistically feasible version of the JCS-ED. Employing exploratory data analysis, the authors find evidence to substantiate the use of a 3-factor, 18-item version of the scale. Subsequent confirmatory analyses provided further verification that the items included in the revised scale were substantively meaningful. Although it is unclear to me why response rates by occupation were unavailable across research sites (presuming the authors were part of the data collection), I am satisfied that other issues have been sufficiently addressed.
---